# Assessing Ride Motion Discomfort Measurement Formulas

Louis T Klauder Jr [1,2]

1   Track Shape and Use LLC, Marshall, MI 49068, USA; lou@klauder.org
2   Louis Klauder, 128 Eastman Ct, Marshall, MI 49068, USA

**Abstract:** This article is about a framework for determining the degree of realism of any given passenger ride motion discomfort measurement formula. After providing some context and reviewing evidence of deficiency in currently popular ride motion discomfort measurement formulas, the article outlines the research program that needs to be carried out in order to establish such a framework. The research begins with gathering recordings of uncomfortable ride motion episodes encountered in a chosen type of passenger transport service. It then has test subjects compare the episodes via a ride motion simulator and adjust their amplitudes pair wise until they cause equal discomfort. It explains how to take the pair wise amplitude adjustments and determine amplitude adjustments that bring all of the motion episode recordings to a common level of discomfort so that they form a normalized set. Then, the lower the scatter of the scores assigned by any given discomfort measurement formula to the members of that set, the more realistic that formula will be for the chosen service.

**Keywords:** vehicle ride motion; discomfort measurement; discomfort formula; motion simulator; cross match





## 1. Introduction

### 1.1. A Way to Assess the Realism of a Ride Motion Discomfort Measurement Formula

This article is generally about the discomfort that seated passengers feel when they are exposed to vibratory and jolting motions while traveling in common carrier vehicles such as planes, trains, and buses. The article takes note of ways that common carriers can make use of ride motion discomfort measurements. It also takes note of some discomfort measurement formulas currently in use for measuring such discomfort.

However, this article neither proposes nor evaluates any particular discomfort measurement formula. Rather, it outlines a research program for creating sets of ride motion recordings all of which cause the same level of discomfort to the "average" passenger. Such a normalized motion recording set can be used to assess the realism of any proposed discomfort measurement formula and thereby can also assist development of realistic formulas. Sections 1–3 of the article provide context and motivation, and Section 4 sets forth the research program for establishing the normalized ride motion recording sets.

This article will sometimes speak in terms of travel by passenger railroad. However, what is presented will also be applicable to measurement of discomfort due to ride motions experienced by seated passengers using other modes of commercial passenger transport. It is general enough to also be used to measure discomfort in occupational situations such as truck driving and equipment operation.

Literature on this subject often refers to "ride comfort". This article refers to "ride discomfort" because that is what passengers sometimes feel and what engineers can try to measure.

This article does not consider the discomfort referred to as motion sickness.

### 1.2. The Motivation for Measurement

Each of the major categories of vehicular passenger transport such as private auto, bus, train, and plane will have its own approach to passenger comfort. In the case of passenger

rail service, if the service is governmentally mandated or has captive ridership it may seem that there is no business reason for measuring the discomfort of the service. As a practical matter, most passenger rail operations are operated with an intent to attract ridership. When passengers have a choice of travel mode, presence or absence of discomfort is a factor in their choice. Thus, a passenger rail service provider is well advised to measure the discomfort to which its patrons are exposed and, if discomfort is identified, to consider whether it would be profitable to take steps to reduce it.

*1.3. Terminology*

This article will use some terms with specific meanings. The ones that will be needed initially are:

- **discomfort** => the subjective discomfort felt by a seated passenger due to vehicle ride motions.
- **discomfort_estimate** => a single number intended to indicate how much **discomfort** an "average" passenger would feel if exposed to a given episode of passenger vehicle seat frame motion. (While some ride motion discomfort research has used measurements of accelerations at seat-surface to passenger-clothing interfaces, measurements on seat frames where they are bolted to floor beams is the most practical choice for ongoing measurements and particularly for those noted in Sections 2.3 and 2.4.
- **discomfort_formula** => a numerical recipe for processing a segment from a digital recording of passenger vehicle seat frame accelerations to obtain a corresponding **discomfort_estimate**.
- **discomfort measurement procedure**, abbreviated as **DMP** => a procedure for recording passenger vehicle seat frame acceleration episodes and using a stated **discomfort_formula** to obtain corresponding **discomfort_estimates**.

## 2. Practical Uses for a Discomfort Measurement Procedure (DMP)

*2.1. A DMP Can Assist in Procurement of New Rail Passenger Vehicles*

The irregularities in the geometry of any railroad track are random and complex and change with time. It is on such track that a new vehicle will be expected to afford good passenger ride quality. The track on which a new vehicle will be required to pass ride quality acceptance tests should be typical of the "roughest" track over which the vehicle is expected to carry passengers. Its exact geometry cannot be stated at the time the vehicle specifications are published, so it is not practical to specify **discomfort_estimate** values that the new vehicles should not exceed. However, the procurement specifications can designate a **discomfort_formula** and an existing vehicle that the new car contractor can study and that is to serve as a standard of comparison. It can then be required that when the new vehicle and the comparison vehicle are both run according to a specified schedule over a specified section of track whose condition will meet the applicable geometric standards, the **discomfort_estimates** generated by the new vehicle may not exceed a stated multiple of the corresponding **discomfort_estimates** generated by the reference vehicle.

*2.2. The Discomfort Formula of a DMP Can Assist in the Design of a New Vehicle*

The concept here is that a **DMP**'s **discomfort_formula** can be employed during the design of a new surface transport passenger vehicle to tune resonant frequencies and suspension damping rates to minimize predicted passenger **discomfort**. Internet searches on phrases such as "rail vehicle ride comfort analysis" can find numerous articles describing such studies. Among them are: (Satari et al., 2022) [1], (Graa et al., 2016) [2], (Herrero 2013) [3], (Dumitriu and Stănică 2021) [4], (Dumitriu and Cruceanu 2017) [5], (He 2003) [6], and (Dižo et al., 2021) [7].

*2.3. A DMP Can Be Used to Help Prioritize Passenger Vehicle Maintenance*

Ride **discomfort** to which passengers are subjected in passenger rail service can arise from defects in vehicle condition. Such defects in particular cars can be discovered when

their **discomfort_estimates** are compared with averages for the fleet. Some general effects of wear and tear over time can be recognized by looking at changes in a fleet average **discomfort_estimate** over time, but in this case possible changes in track condition with time must also be considered. In passenger services that do not employ conductors this use should be cost-effective. In a service where ride quality anomalies are systematically reported by conductors the cost-effectiveness of this use is open to question but may still be positive.

*2.4. A DMP Can Be Used to Help Prioritize Track Maintenance*

Ride **discomfort** to which railroad passengers are subjected in service can arise from local track defects. Such defects can be discovered when **discomfort_estimates** are looked at as a function of track location. This application can provide a beneficial supplement to the basic track maintenance procedures that are in place to ensure safety, promote efficiency, and satisfy regulatory requirements.

### 3. Discomfort _Formulas in Use and Their Inadequacies

*3.1. Discomfort _Formulas for Single Axis Pure Sinusoidal Motions*

The simplest kind of nonuniform motion is sinusoidal motion along or about a single axis. Procedures have been carried out to determine how **discomfort** felt by passengers exposed to such simple motions varies with frequency and with the choice of the axis of vibration or rotation.

The best-known reference where such results are set forth is the International Standards Organization's ISO 2631-1997 titled "Mechanical vibration and shock-Evaluation of human exposure to whole-body vibration", Part-1, "General requirements" (ISO 2631-1 1997) [8], which will be referred to as just ISO 2631. This standard presents formulas and frequency dependent weighting factors for estimating the discomfort experienced by people exposed to single frequency, single axis sinusoidal accelerations. It also presents suggestions for estimating discomfort caused by more complex ride motions.

*3.2. Discomfort Formulas for Complex Motions*

Articles in the field of vehicle ride motion **discomfort** research commonly report estimation of relative **discomfort** using formulas defined in one or more of the following publications:

(A) ISO 2631 Part 1 1997 [8], noted above. That standard has several other parts among which part 4 (ISO2632-4 2001) [9] gives recommendations for estimation of **discomfort** engendered by ride motions of passenger rail vehicles.

(B) British Standards Institution, BS 6841, Measurement and evaluation of human exposure to whole-body mechanical vibration and repeated shock (BS 6841 1987) [10]. This elaborates on ISO 2631 with a British perspective.

(C) BS EN 12299:2009, Railway applications—Ride comfort for passengers. Measurement and evaluation (BS EN 12299:2009) [11] is the English language version of a European standard. EN 12299 generally follows the recommendations in ISO 2631 but supplements them with additional detailed advice about how to record and process data. Illustrations of processing called for in EN 12299 can be seen in slides of a talk given in 2016 by Bjorn Kufver (Kufver, B. 2016) [12].

(D) Sperling, "Contribution to the evaluation of ride comfort in rail vehicles" (Sperling, E. 1956) [13]. This publication appeared before the others, is different in detail but similar in approach, and remains popular in a number of countries.

In the above list of major standards, B and C assume and reproduce the basic material of A and add recommendations about procedures for gathering and processing data. D in the list is analogous to A in that it is based on spectral decomposition of motion signals.

Copies of the first three of these standards are offered for sale at fairly high prices. A reader who does not have access to the standards themselves can find summaries of

their basic formulas in several of the references cited below such as (Wawryszczuk et al., 2023) [14] and (Dumitriu and Leu 2018) [15].

*3.3. How Current Discomfort _Formulas Conceptualize Ride Motion*

The above four publications all approach ride motion **discomfort** under the influence of two main ideas.

The first is that when dealing with oscillatory phenomena it is customary to resolve them into their sinusoidal Fourier components. It is relatively simple to expose test subjects to sinusoidal motions and to record their judgments about the degree of **discomfort** that those oscillatory motions engender. Such results are well attested, stable over time, and widely accepted. Curves documenting the way that human sensitivity to sinusoidal motions varies with frequency for each choice of axis of translation or rotation are documented in ISO 2631 and in standards that are based thereon.

The second is the assumption that human response to an oscillatory motion as a whole can be satisfactorily estimated by the sum of the responses that would be engendered by each of the suitably weighted spectral components of that motion. The components are typically grouped into 1/3rd octave bands.

In line with those two ideas, these four standards begin their evaluation of a recorded ride motion by Fourier analyzing its acceleration signals into frequency bands. They then multiply the amplitude of each spectral component by a frequency and axis dependent weighting factor, raise each weighted spectral component to a stated exponent, and sum the results for all the spectral components.

As an example, ISO 2631-1997 Part-1 clause 6.1 calls for calculation for each axis of vibratory motion of a basic single axis frequency weighted RMS acceleration (FWRA) measure defined as

$$a_w = \left[ \frac{1}{T} \int_0^T a_w^2(t) dt \right]^{1/2}$$

where $a_w(t)$ is a modified time dependent linear or rotational acceleration wave form constructed from the Fourier components of the acceleration recording by multiplying each component by a weight appropriate for its axis and frequency (When, as here, the square of the signal is being averaged, the result would be calculated in the frequency domain by summing the squares of the weighted Fourier components to save the step of converting from the frequency domain back to the time domain). Then RMS values of relevant single axis measures can serve as composite measures of **discomfort** for complex ride motions.

At the same time, some of those standards suggest alternate formulas for motions that are far from sinusoidal. For instance, ISO 2631 Clause 6.3 begins with:

"In cases where the basic evaluation method may underestimate the effects of vibration (high crest factors occasional shocks, transient vibration), one of the alternative measures described below should also be determined—the running r.m.s. or the fourth power vibration dose value". Those alternate methods are given in clauses 6.3.1 and 6.3.2.

Clause 6.3.1 reads:

"6.3.1 The running r.m.s. method.

The running r.m.s. evaluation method takes into account occasional shocks and transient vibration by use of a short integration time constant. The vibration magnitude is defined as a maximum transient vibration value (MTW), given as the maximum in time of $a_w(t_0)$, defined by:

$$a_w(t_0) = \left\{ \frac{1}{\tau} \int_{t_0-\tau}^{t_0} [a_w(t)]^2 dt \right\}^{1/2}$$

where

$a_w(t)$ is the instantaneous frequency-weighted acceleration;

$\tau$ is the integration time for running averaging;

$t$ is the time (integration variable);

$t_0$ is the time of observation (instantaneous time)".

$\cdots$ (omitting a few lines about an approximation method)

The maximum transient vibration value, MTW, is defined as

$MTW = max[a_w(t_0)]$

i.e., the highest magnitude of $a_w(t_0)$ read during the measurement period (T in 6.1).

It is recommended to use $\tau = 1$ *sec* in measuring MTW (corresponding to an integration time constant, "slow", in sound level meters)."

Clause 6.3.2 begins:

"6.3.2 The fourth power vibration dose method

The fourth power vibration dose method is more sensitive to peaks than the basic evaluation method by using the fourth power instead of the second power of the acceleration time history as the basis for averaging. The fourth power vibration dose value (VDV) in meters per second to the power 1.75 ($m/s^{1.75}$), or in radians per second to the power 1.75 ($rad/s^{1.75}$), is defined as:

$$VDV = \left\{ \int_0^T [a_w(t)]^4 dt \right\}^{1/4}$$

where

$a_w(t)$ is the instantaneous frequency-weighted acceleration;

$T$ is the duration of measurement (see 6.1)".

The method proposed by Sperling is employed in (Dumitriu and Stănică 2021) [4]. There the formula for its measure of **discomfort** due to the vertical component of ride motion is expressed as

$$W_z = \left[ \int_{0.5}^{30} a^3(f) \, B^3(f) \, df \right]^{1/10}$$

where $f$ denotes frequency, $a(f)$ is the Fourier transform of the vertical acceleration, and $B(f)$ represents Sperling's estimate of the way that human discomfort due to sinusoidal vertical motion varies with frequency. It has a peak at about 5.2 Hz and falls sharply to either side of that peak.

The above formulas illustrate the Fourier decomposition employed in the principal widely used methods of estimating ride motion **discomfort**. The running RMS, FWRA, and VDV measures appear to be based on acceleration as a function of time, but they use an artificial frequency weighted acceleration rather than the acceleration actually experienced by passengers. As the running RMS and FWRA measures average the square of the acceleration in the time domain, their results are identical to corresponding frequency domain integrals indicating that they discard acceleration wave form information and keep only energy information. In contrast, what passengers experience is the ensemble of acceleration wave forms.

### 3.4. Evidence That Currently Used Discomfort Formulas Can Be Unrealistic

Papers (Araújo et al., 2010) [16], (Kaneko, Hagiwara, and Maeda 2005) [17], (Maeda and Mansfield 2006) [18], and (Maeda, Mansfield, and Shibata 2008) [19] describe investigations and comparison of results with the recommendations in ISO 2631 and conclude that those recommendations do not correlate very well with perceived discomfort caused by some motions that are not single axis and single frequency. (Maeda, Mansfield, and Shibata 2008) [19] and (Mansfield and Maeda 2011) [20] also report that subjective responses to

broad-band random ride motions correlate better with the RMS type measures defined in ISO 2631 if the frequency dependent spectral weightings recommended therein are omitted.

Plewa et al. (2012) [21] reports that for seat accelerations experienced by operators of some forestry and mining vehicles the levels of discomfort reported by the operators showed almost no relationship to the discomfort scores calculated according to ISO 2631. The ride motions of that study are more abrupt and more uncomfortable than those normally encountered in even the least comfortable passenger rail vehicles.

## 4. A Research Program for Forming Sets of Equal Discomfort Motion Recordings

### 4.1. A Logical Approach

The papers referenced in Section 3.4 show that the ride measures commonly in use yield results that disagree more or less with passenger perceptions of **discomfort** due to ride motions encountered in daily life. In published investigations into how to measure ride quality, the general approach has been to consider one or more published or proposed **discomfort_formulas** and to compare its or their scorings of laboratory or revenue service ride motions with test subject verbal scorings of those motions.

In contrast to that traditional approach to studying **discomfort_formulas**, it would be both more logical and more effective to assemble a collection of digital recordings of diverse episodes of uncomfortable real life ride motions and to adjust the amplitudes of those recordings so that, on average, test subjects considered them all equally uncomfortable. With such an equal-**discomfort** motion recording collection available, the realism of any prospective **discomfort_formula** could easily be determined using just a personal computer. All that would be required would be to apply the **discomfort_formula** to each ride of the equal-**discomfort** collection and calculate the scatter of the resulting scores. The smaller the scatter, the more realistic the **discomfort_formula**. If the **discomfort_formula** had adjustable parameters, they could easily be optimized to minimize the scatter of the scores on that collection.

In order for this research program to be carried out there needs to be a procedure for bringing an initially gathered set of motion episode recordings to a common level of **discomfort** as perceived on average by test subjects. The procedure proposed herein for that purpose has two steps.

The first step compares pairs of motion recordings one pairing at a time. It makes use of a motion simulating shaker table with passenger seating attached. Test subjects are instructed that when seated thereon, they will be exposed alternately to two motion recordings, and that while being so exposed they are to use a control knob to adjust the amplitude of one motion until they feel that the **discomfort** of that motion matches the **discomfort** caused by the other one. The selected gain settings are recorded. When the comparisons are all finished, then for each pairing the gain values selected by subjects for that pair are averaged.

The second step is a mathematical procedure that takes those averaged individual pairing gain adjustment factors and derives for each recording a gain factor that will bring that recording to a common level of **discomfort**. Carrying out those two steps will be referred to as a "round" of recording amplitude adjustment.

As will be explained below, the second step assumes that perceived passenger **discomfort** will vary linearly with motion amplitude scaling. That is expected to be true to good approximation only for "small" changes in motion amplitude. The originally gathered ride motion recordings will normally engender substantially different levels of **discomfort**. Therefore use will be made of some existing **discomfort_formula** such as the RMS value or one of those listed in Section 3.2 to adjust the amplitudes of the recordings to bring them to a common level of **discomfort** (as judged by that existing **discomfort_formula**) before their use in the first round of comparisons.

We expect that the results of a second round that starts with recordings whose amplitudes have been adjusted toward **discomfort** equality based on the results from the first round will be good enough for practical purposes. In an initial performance of this

procedure it will be important to conduct a third round to test that assumption. It could turn out that three rounds are generally desirable or even that with prior adjustment a single round may generally suffice.

As far as the author is aware the only paper that has proposed this approach is the 1975 paper (Klauder and Clevenson 1975) [22] which described work done to carry out and process one round of comparisons. That paper did not get much attention, perhaps because it was in the proceedings of a symposium rather than in a journal, was before the days of the internet, and started out with some non-essential theory that might have discouraged further reading. The goal of this paper is to give the core of that paper a second hearing and hopefully persuade the vehicle ride quality community of its utility.

### 4.2. Additional Terminology

Use will be made of some additional terms as follows:

- **in_sample** => a multi-channel digital recording of a short episode of passenger vehicle seat base accelerations that causes annoying **discomfort**. **In_samples** are selected from original field recordings made on operating revenue vehicles and should be diverse representative examples of the most annoying acceleration episodes encountered during revenue operations. They are used, with amplitudes suitably scaled, to drive the motion simulator. Before the first comparison round they are scaled toward a common level of **discomfort** using some pre-existing **discomfort_formula**. At the end of each round they are scaled toward equal **discomfort** based on test subject adjustments recorded during that round.
- **sample_set** => a collection of **in_samples**.
- **out_sample** => a multi-channel digital recording of the accelerations of the base of a seat on the motion simulator while the simulator is driven by the corresponding **in_sample**. This recording is made before the **in_sample's** amplitude is raised or lowered to insure that the test subject will need to change it to achieve perceived equality of discomfort, and thus also before the test subject begins to adjust it.
- **sample** => the sample identification shared by the **in_sample** and **out_sample** manifestations thereof.
- **normalized_set** => the set of **out_samples** that emerge from a comparison round with their amplitudes scaled toward a common level of **discomfort** based on the results of that round.
- **scatter** => a value such as the dispersion or mean absolute deviation indicating the extent to which the **discomfort_estimates** obtained by applying a **discomfort_formula** to the **out_samples** of a **normalized_set** differ from their average.

Summarizing two key points in the above definitions: after all the gain factors for a round have been computed, the **in_samples** are all scaled toward a common level of **discomfort** so that they will be ready to be used as inputs in the next round (if any), and the **out_samples** are all scaled toward a common level of **discomfort** so that they will constitute a **normalized_set**.

### 4.3. Some Diagrams Illustrating the Progression of Steps Described below

Figure 1 that follows presents some flow diagrams that may help the reader keep track of the procedural steps set forth in the sections that follow.

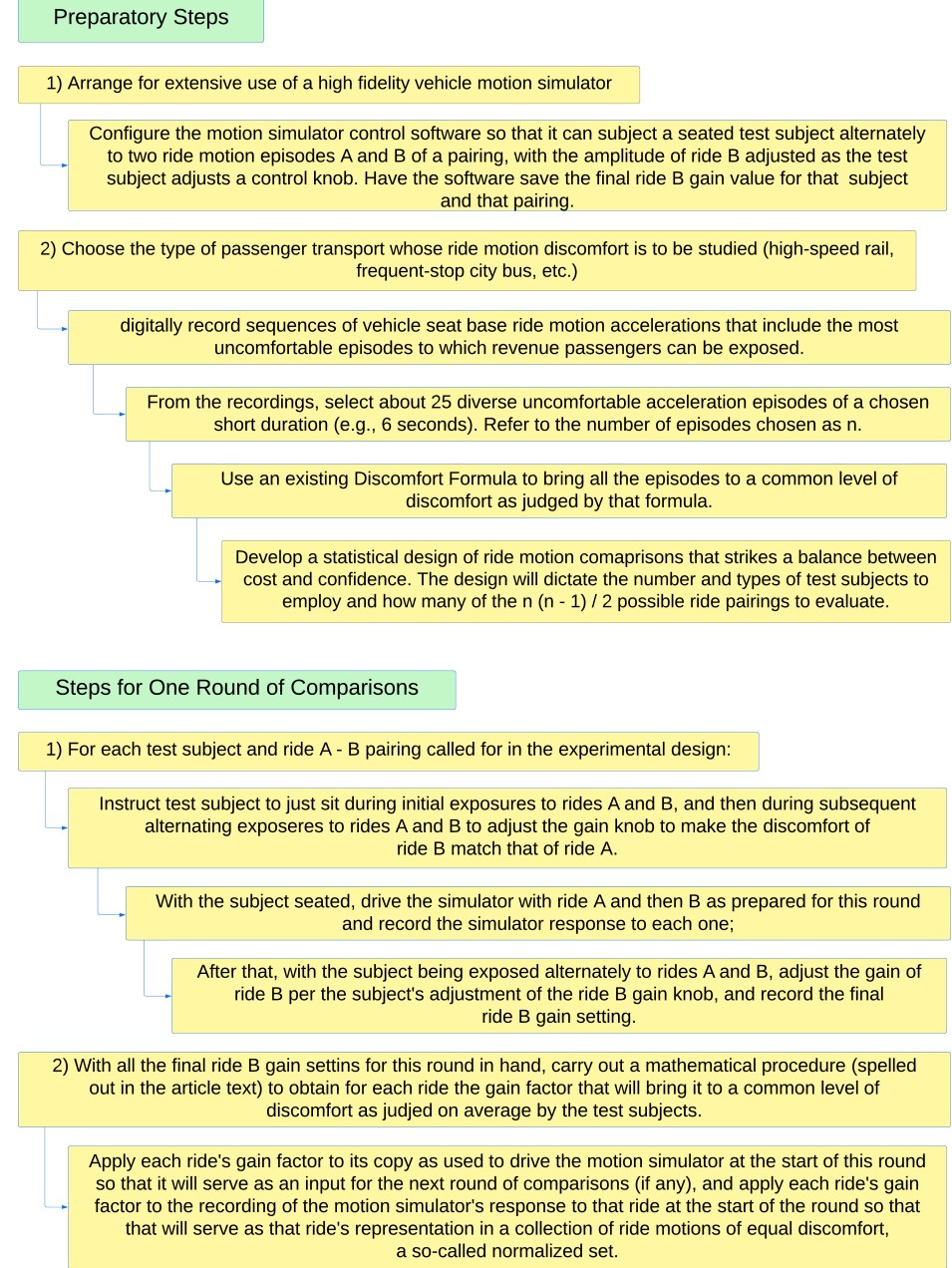

**Figure 1.** Flow charts depicting steps of proposed research program.

*4.4. Assembling a Sample Set*

This subsection discusses the initial gathering of revenue service ride motion recordings.

The first step is to select the revenue services on which to make field recordings. The project will presumably aim to establish a **normalized_set** for qualifying or developing a **discomfort_formula** for some specific type of passenger service. It might then seem logical to limit field recording to revenue services of that type. However, if additional recordings are made on other types of service, then it might be possible to show that a **discomfort_formula** optimized for the target type of service would also give realistic scores to disturbing ride motions found on other types of service.

When accelerations are reproduced by the motion simulator they are realized at its floor to which seat bases are attached. Thus the accelerations that need to be recorded to characterize revenue service ride motions are those of seat frame bases where they are attached to the vehicle floor. That way a seat on the motion simulator will be moved in

almost the same way as the passenger car seat on whose frame base the accelerations were recorded.

When contemplating the possibility that a **discomfort_formula** developed for one type of revenue service might give equally realistic scores when used with another type of service, account must be taken of the cushioning role of seating upholstery. Accelerations experienced by passengers seated on soft upholstery in long distance services will generally be lower than the associated seat base accelerations. In contrast, passengers on hard plastic seating in rapid transit services will feel the full brunt of seat frame accelerations. It would therefore seem impossible in principle for a **discomfort_formula** optimized for a service with hard seating to be the same as one optimized for a service with soft seating.

Related to the foregoing, it is important that the seating used on the motion simulator have cushioning and seat frame mechanical resonances that are representative of the target type of revenue service.

The transient and oscillatory motion signals that can be recorded without difficulty are the linear accelerations in the vertical, lateral, and longitudinal directions and the angular yaw, pitch, and roll accelerations. As a practical matter, when dealing with long wheel base vehicles typically used for passenger rail service, the yaw and pitch accelerations are usually ignored leaving just the linear and roll accelerations. The longitudinal and roll accelerations may sometimes also be ignored.

The linear acceleration signals are typically band pass filtered to retain the spectral content between 0.4 or 0.5 Hz and 80 or 100 Hz with filtering that minimizes wave form distortion. This filtering can be incorporated in the recording process or applied after recordings have been gathered.

It is desirable for the the field recording instrumentation to include a voice channel and a channel for documenting the vehicle's location. On the voice channel the person conducting the recording can describe notable motion disturbances as they are encountered.

The initial field recordings of acceleration signals are likely to have durations in the range of 5 min to two hours. From the field recordings it is necessary to select episodes lasting for a chosen duration between 5 and 11 s that will constitute the **in_samples**. They should include examples of as many different types of ride disturbance as practical. The focus should be on selecting the most uncomfortable episodes. Having **in_sample** durations in the indicated range is to facilitate A-B comparisons using the motion simulator. The amount of disturbing motion should be relatively constant throughout each **in_sample**. Episodes of short duration abrupt disturbance might be duplicated to form **in_samples** with the chosen length and relatively steady **discomfort**. Field recording voice commentary, location information, and a computer display of acceleration wave forms can help the person deciding where on a field recording each **in_sample** should begin.

Note that because the proposed procedure employs **samples** with some uniform short duration, it is not designed to allow assessment of **discomfort_formulas** that attempt to predict the way that perceived **discomfort** depends on the duration of exposure to persistent uncomfortable ride motions due to factors such as unbalanced wheels and failed shock absorbers of buses or truck hunting of rail road cars.

*4.5. Step One in a Comparison Round: A-B Comparisons via the Simulator*

We will now say a little more about step one of the process for converting a **sample_set** into an equal-discomfort **normalized_set**. Much of the material of this section and the next is adapted from (Klauder and Clevenson 1975) [22].

The author's thinking in this area was informed by a 1970 paper by C. Ashley (Ashley 1970) [23]. Ashley used two side-by-side "shaker tables" as motion simulators and had test subjects stand alternately on:

(A) a table driven by a broad band random signal and

(B) a table driven by a sinusoidal signal.

As subjects experienced alternately the random motion and the sinusoidal motion they adjusted the amplitude of one of the signals to get the **discomfort** of the two motions

to be the same. This was done for a sequence of sinusoidal frequencies in two stages. Ashley's paper gives details. Ashley's procedure constitutes a significant improvement over procedures which seek to have subjects verbally compare ride motions which differ in **discomfort**, and it is a model for that aspect of the procedure advocated here. Ashley referred to this technique of data collection as "cross matching".

Some of the concepts employed in this section are mentioned in (Maeda and Mansfield 2006) [18]. Two later papers that describe A-B comparisons are (Strandemar 2005) [24] and (Zong et al., 2000) [25].

The comparison procedure of the first step requires use of a capable vehicle ride motion simulator on whose platform is mounted a seat module. The seat module should be representative of the seating used in the revenue service for which the **normalized_set** of motion samples is being prepared.

Electronic means need to be in place to:

- feed the motion simulator alternately with the signals of **samples** A and B of the current pairing, varying which of the two is presented first.
- allow test subjects to adjust the amplitude of **sample** B. It will probably be prudent to have each subject go through two comparison sequences for each pairing with the amplitude of **sample** B scaled to start out alternately more and less uncomfortable than **sample** A.
- illuminate a sign to keep the subjects aware of which **sample** they are currently experiencing.

The test subjects are to be instructed that as they are exposed alternately to two motion **samples**, A and B, they are to adjust the gain of **sample** B to make its **discomfort** match that of **sample** A. In the testing reported in (Klauder and Clevenson 1975) [22] each **sample** was presented for 10 s, and there was a 2 s pause between alternate **samples**.

*4.6. Step Two in a Comparison Round: Calculating the Gain Factors*

Let $n$ denote the number of **samples** of the **sample_set**. Label the **samples** from 1 to $n$ in an order that is randomized afresh for each round. Let $g_{ij}$ denote the gain value that when applied to the amplitude of **sample** $j$ makes its **discomfort** equal to that of **sample** $i$. (That is inverse to the definition used in reference [22]) Moreover, define $g_{ij}$ as a true gain factor that is not effected by inconsistencies in test subject responses. $g_{ii}$ always equals 1 and would be of no interest except that $g_{nn} = 1$ plays a role below in the formula for the geometric mean of all of the gain factors.

For practical purposes we will assume that $g_{ji} = 1/g_{ij}$. That will not be strictly true because human response to mechanical stimulation is not linear. What that means in this context is that (on average over test subjects) the factor by which a test subject changes the amplitude of motion A to achieve equal discomfort will not be exactly one over the factor by which he or she would change the amplitude of B to achieve equal discomfort. However, as the discomforts of the motions are brought closer to equality the amplitude adjustment factors will approach unity so that that equation will become increasingly valid from round to round. This problem of non-linearity is the reason for the need to conduct at least two rounds of A-B comparisons on the motion simulator and to use results from one round to bring all the **in_samples** closer to a common level of **discomfort** before carrying out the next round.

In light of that approximate equality we can cut the number of comparisons in half by making only those comparisons for which $i > j$. We will then be dealing with the $n(n-1)/2$ $g_{ij}$ values for which $i > j$. We note next that the set of $g_{ij}$ factors possesses only $(n-1)$ degrees of freedom; namely all the $g_{ij}$ values can be obtained from the $(n-1)$ values $g_{n1}, g_{n2}, g_{n,n-1}$ via the relations, $g_{ij} = g_{in}g_{nj} = g_{nj}/g_{ni}$.

Coming back to the human subject responses that step one will have yielded, let $r_{ij}$ denote the average of the gains assigned by test subjects to **sample** $j$ to make its **discomfort** match that of **sample** $i$ during a given comparison round. All the $r_{ij}$ values will have $i > j$. We want to find the set of $g_{ij}$ values that provides the best fit to the empirical $r_{ij}$

values. The variables to be determined are $g_{n1}, g_{n2}, \cdots g_{n,n-1}$, which we will abbreviate as $g_1, g_2, \cdots g_{n-1}$. ($g_n = 1$ by definition.) We find the best fit by minimizing an error function that measures the extent to which the $g_i$ values fail to be consistent with the $r_{ij}$ values. For the error function to be minimized we take

$$E = \sum_{i>j}' \left[ \frac{g_{ij}}{r_{ij}} - 1 \right]^2 = \sum_{i>j}' \left[ \frac{g_{in}g_{nj}}{r_{ij}} - 1 \right]^2 = \sum_{i>j}' \left[ \frac{g_{nj}}{g_{ni}r_{ij}} - 1 \right]^2 = \sum_{i>j}' \left[ \frac{g_j}{g_i r_{ij}} - 1 \right]^2$$

where the prime over the summation symbol indicates here that a given $(ij)$ pair is not to be included in the sum if the corresponding $r_{ij}$ happened not to be measured.

The $g_i$ values which minimize $E$ are found with the help of a simple computer code which uses Newton's method and iterates until the partial derivatives, $\frac{d E}{d g_i}$, are all close to zero.

With the $g_{ni} = g_i$ values in hand, each of the **samples** can have its **discomfort** level adjusted to match that of **sample** $n$. To accomplish that one simply multiplies the amplitude of **sample** $i$ by $g_i$. The **samples** as thus adjusted will constitute a **normalized_set**.

The geometric mean of the factors by which the **samples** of the foregoing **normalized_set** will have had their amplitudes adjusted is $g_{mean} = [g_1 g_2 \cdots g_n]^{1/n}$. The **normalized_set** whose members have a comfort level matching the mean of the comfort levels of the **samples** of the original **sample_set** is obtained by multiplying each original **sample** $i$ not by $g_i$ but rather by $g_i / g_{mean}$.

As noted above, determination of the $g_i$ values should be accomplished by carrying out two or more rounds of A-B comparisons with each round serving to bring all of the ride samples closer to a common level of **discomfort**. Adjustments will become smaller as rounds progress, and successive rounds will have an averaging effect that will reduce the impact of inconsistencies due to human variability.

In order for the research proposed herein to be conducted, there will need to be substantial funding, hosting by a well equipped and staffed research laboratory that operates a high-fidelity motion simulator of sufficient capacity, and the participation of professional researchers with competence in psychophysics, test subject management, statistical design and analysis, simulator operation, data recording, and data processing. Such a team would revise and elaborate the procedural details proposed herein. As an example, the statistician and psychophysicist would need to work together to decide how many test subjects to employ and how to go about enlisting them.

## 5. Conclusions: A Discomfort _Formula Can Be Accurately Evaluated

Given a **normalized_set** and a prospective **discomfort_formula** the number that indicates the realism of the **discomfort_formula** with respect to the type of revenue service represented by the **normalized_set** is the **scatter** of the **discomfort_estimates** obtained when the **discomfort_formula** is applied to each of the **samples** of the **normalized_set**. The lower the **scatter**, the more realistic the **discomfort_formula**. As noted previously, that same procedure can be used to optimize any adjustable parameters present in a prospective **discomfort_formula**.

## 6. Discussion

Reference (Klauder and Clevenson 1975) [22] includes an example of use of the **normalized_set** described therein to optimize one hypothetical **discomfort_formula** that included 14 adjustable parameters. Even when optimized that particular **discomfort_formula** did not give realistic results. A little subsequent exploration found that an exceedance type **discomfort_formula** (Catherines, Clevenson, and Scholl 1972) [26], (Vinje 1972) [27] gave very consistent scores to those **samples** of the **normalized_set** that represented motion episodes recorded in revenue services. However, it did poorly on the one **sample** that consisted of an artificial sinusoidal motion. That subsequent exploration was not published,

and unfortunately, organizational and computer resource changes that occurred shortly after that work was done lead to loss of the underlying **normalized_set** data.

Hopefully some research group will take up the procedure described herein so that concrete progress can be made. It would be very helpful if such a program yielded open access downloadable **normalized_sets** representative of the major types of commercial passenger ground transport. Such **normalized_sets** would constitute a basis for development and validation of realistic **discomfort_formulas**.

**Funding:** This research received no external funding.

**Data Availability Statement:** No new data were created or analyzed in this study. Data sharing is not applicable to this article.

**Acknowledgments:** The author thanks Setsuo Maeda for his encouragement and thanks two peer reviewers for their helpful comments.

**Conflicts of Interest:** The author declares no conflicts of interest.

## Abbreviations

The following abbreviations are used in this manuscript:

DMP discomfort measurement procedure
FWRA frequency weighted RMS acceleration
ISO International Standards Organization
MDPI Multidisciplinary Digital Publishing Institute
MTW maximum transient vibration value
RMS root mean square (i.e., square root of average of squares)
VDV fourth power vibration dose value

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
