# Peer review of "Assessing Ride Motion Discomfort Measurement Formulas"

_vibration, doi:10.3390/vibration7020022_

Round 1

Reviewer 1 Report

Comments and Suggestions for Authors

There are 3 main issues with this manuscript (see below).

1- Even though this manuscript revolves around passenger's perception of ride comfort/discomfort, A-B comparisons, etc., there is no mention of anything related to psychophysics theory/protocols in the manuscript itself - This is unsettling. How can it be?

2- Some parts of the manuscript are unintelligible to me. For instance, despite my best effort, I cannot comprehend most of what is presented on page 7. Please make it easier for the reader to understand/assess your research/manuscript. 

3- The author relies on acceleration alone for comfort/discomfort assessments. Is acceleration a reliable metric? How about transmitted power or energy? I invite the author to read the following articles :

** Drouet, JM., Covill, D., Leroux, M. et al. On metrics to assess road bicycle dynamic comfort during impacts. Sports Eng 25, 1 (2022). https://doi.org/10.1007/s12283-021-00366-x

** Ayachi, F. S., Drouet, J.-M., Champoux, Y., & Guastavino, C. (2018). Perceptual Thresholds for Vibration Transmitted to Road Cyclists. Human Factors60(6), 844-854. https://doi.org/10.1177/0018720818780107

Comments on the Quality of English Language

Please check manuscript for typos.

Reviewer 2 Report

Comments and Suggestions for Authors

This study aimed to propose a method to evaluate and develop discomfort measurement formulas by using recordings of uncomfortable ride motions in public transport and comparing them. A new framework for evaluating the discomfort formulas has been described and explained.    The paper tried to explore an alternative framework for discomfort formulas, which is interesting and could be beneficial for practical evaluation.

The motivation of the research has been well justified in sections 1-3. The different discomfort formulas have been mentioned and explained in section 3. This is an important driving point of the current paper, please give a bit more detail to summarise the difference between the listed four methods, especially on the continuous motion and shock motion, and explain the limitations of the current evaluation formulas.

The method proposed in this paper is a standard psychophysical method (method of adjustment). One of the main features of this method is the opportunity afforded to a subject to control the changes in the stimulus, however, this can be a disadvantage and the resulting data may not be as reliable. The criterion for matching is important - what one asks the subject to do and what the subject does are not at all the same thing. Please give a bit more details on the instructions you given to the subjects. Any practice needed to achieve reliable results.

One main issue with discomfort evaluation is non-linearity. For vibration discomforts, the low-range magnitude would cause different responses compared with the high-level magnitude motion. This brings in one question, how do you set up your sample vibrations (reference point), and at what intervals? Please add comments on this.

The other important factor to consider is the duration of the motion. How do you address a relatively short-term (lab) discomfort with a long-journey discomfort (a combination of bot static and dynamic seating discomfort)?

The last point is the selection of sample sizes (how many subjects) is required to be representative. Please add comments on this.

Other psychophysical methods (e.g. magnitude estimation) are commonly used for lab discomfort research, worth looking into and comparing with your proposed method.

Round 2

Reviewer 2 Report

Comments and Suggestions for Authors

Comments have been addressed in the revised draft. 

Author Response

The initial comments and suggestions offered by Reviewer 2 were very helpful and demonstrated an expert grasp of the subject matter. He has graciously accepted the changes made in response thereto and has not suggested any further revisions. (The editor for the journal has made some suggestions of a formal nature, and those are addressed in a separate response to the editor.)